# Quantitative In-Depth Transcriptome Analysis Implicates Peritoneal Macrophages as Important Players in the Complement and Coagulation Systems

**DOI:** 10.3390/ijms23031185

**Published:** 2022-01-21

**Authors:** Aida Paivandy, Srinivas Akula, Sandra Lara, Zhirong Fu, Anna-Karin Olsson, Sandra Kleinau, Gunnar Pejler, Lars Hellman

**Affiliations:** 1Department of Medical Biochemistry and Microbiology, Uppsala University, The Biomedical Center, SE-751 23 Uppsala, Sweden; aida.paivandy@imbim.uu.se (A.P.); Anna-Karin.Olsson@imbim.uu.se (A.-K.O.); gunnar.pejler@imbim.uu.se (G.P.); 2Department of Cell and Molecular Biology, Uppsala University, The Biomedical Center, SE-751 24 Uppsala, Sweden; srinivas.akula@icm.uu.se (S.A.); sandra.lara@icm.uu.se (S.L.); fuzhirong.zju@gmail.com (Z.F.); sandra.kleinau@icm.uu.se (S.K.)

**Keywords:** macrophage, monocyte, transcriptome, mRNA, liver, complement system, coagulation system

## Abstract

To obtain a more detailed picture of macrophage (MΦ) biology, in the current study, we analyzed the transcriptome of mouse peritoneal MΦs by RNA-seq and PCR-based transcriptomics. The results show that peritoneal MΦs, based on mRNA content, under non-inflammatory conditions produce large amounts of a number of antimicrobial proteins such as lysozyme and several complement components. They were also found to be potent producers of several chemokines, including platelet factor 4 (PF4), Ccl6, Ccl9, Cxcl13, and Ccl24, and to express high levels of both TGF-β1 and TGF-β2. The liver is considered to be the main producer of most complement and coagulation components. However, we can now show that MΦs are also important sources of such compounds including C1qA, C1qB, C1qC, properdin, C4a, factor H, ficolin, and coagulation factor FV. In addition, FX, FVII, and complement factor B were expressed by the MΦs, altogether indicating that MΦs are important local players in both the complement and coagulation systems. For comparison, we analyzed human peripheral blood monocytes. We show that the human monocytes shared many characteristics with the mouse peritoneal MΦs but that there were also many major differences. Similar to the mouse peritoneal MΦs, the most highly expressed transcript in the monocytes was lysozyme, and high levels of both properdin and ficolin were observed. However, with regard to connective tissue components, such as fibronectin, lubricin, syndecan 3, and extracellular matrix protein 1, which were highly expressed by the peritoneal MΦs, the monocytes almost totally lacked transcripts. In contrast, monocytes expressed high levels of MHC Class II, whereas the peritoneal MΦs showed very low levels of these antigen-presenting molecules. Altogether, the present study provides a novel view of the phenotype of the major MΦ subpopulation in the mouse peritoneum and the large peritoneal MΦs and places the transcriptome profile of the peritoneal MΦs in a broader context, including a comparison of the peritoneal MΦ transcriptome with that of human peripheral blood monocytes and the liver.

## 1. Introduction

Macrophages (MΦs) were likely the first immune cells to appear during eukaryote evolution, and MΦ-like cells have been found in almost all multicellular organisms. They are present in all mammalian organs, represented by microglial cells in the brain, Kupffer cells in the liver, osteoclasts in the bone, alveolar MΦs in the lung, synovial A cells in the joints, kidney MΦs, gingival MΦs surrounding the teeth, peritoneal MΦs, intestinal MΦs as well as several MΦ types in the lymph nodes and spleen, including marginal zone, metallophilic and red pulp MΦs [1,2]. 

For many years, MΦs were thought to originate exclusively from blood monocytes, which after entering local tissues and under the influence of the local environment (including cell–cell contacts and different soluble factors) develop into MΦs of different phenotypes. However, it has recently been shown that many MΦ subpopulations primarily originate from an early wave of MΦs emanating from the yolk sac, and that these cells can increase in numbers by local proliferation [3,4,5,6]. Microglial cells seem to almost exclusively originate from this early wave of MΦ colonization of the brain [7,8]. This is also the case for the majority of the Kupffer cells of the liver, the alveolar macrophages, and the peritoneal MΦs [5]. In contrast, the intestinal MΦs seem almost exclusively to originate from blood monocytes [9]. Interestingly, although the majority of these MΦ subpopulations can self-renew under normal physiological conditions, almost all of them can be replaced by blood monocytes following experimental depletion [10,11,12,13,14]. However, it is not known how long-lived such monocyte-derived macrophages are in the respective tissues or whether they can self-renew within the tissue (similar to the yolk-sac-derived MΦs). 

MΦs have been found to have a prominent role in many human diseases such as cancer, atherosclerosis, bone healing, scar formation, and sensitivity to infection. In addition, malfunction of lung MΦs may result in aberrant accumulation of lung surfactants, thereby causing reduced lung function. In tumors, MΦs can constitute up to 50% of the total cell number, and in such settings, there is a balance between the two major subpopulations of MΦ, defined as inflammatory M1 MΦs and immunosuppressive M2 MΦs [1]. In the presence of M1 MΦs, tumors have a lower chance of survival, whereas if the tumor can trigger a switch from M1 to M2 MΦs, tumors have an increased probability of avoiding an immune attack [1]. MΦs are also key players in the development of adaptive immune responses mediated by B and T lymphocytes. In the absence of inflammatory signals, the levels of MHC class II on MΦs are low but can increase upon engagement of pattern recognition receptors. Under such circumstances, MΦs will switch from a tissue homeostasis mode to an inflammatory mode and will thereby upregulate the expression of MHC class II and several of the early inflammatory cytokines including IL-1, IL-6, TNF-α, and IL-12 [15,16,17,18]. 

Although numerous studies have addressed the transcriptome and proteome of MΦs found in different tissues, under various inflammatory conditions and in various diseases, there are, to our knowledge, no in-depth quantitative analyses of MΦ transcriptome in healthy tissues have been performed, except for lineage tracing (by single cell analysis) of top-expressed or lineage-related transcripts. Notably, due to the very low number of transcripts recovered from each single cell vs. a purified cell fraction, the variability in expression levels originating from the single cell analysis will therefore be high, which may also reflect technical difficulties in obtaining good coverage of all transcripts from a single cell. Together, this will limit the value of such results in determining accurate expression levels within a given cell population. Nevertheless, the large number of such studies that have been performed have resulted in a roadmap of the major phenotypic differences between different MΦ/monocyte and dendritic cell populations [19]. 

To obtain a quantitative view of the phenotype of different MΦ subpopulations and to evaluate the biological significance of their expressed proteins, we here analyzed the transcriptome of the major subpopulation of peritoneal MΦ in the mouse, the large peritoneal MΦ from Balb/c mice. Our study builds on previous studies but contrasts from them by providing a more detailed and quantitative view of the peritoneal MΦ transcriptome under steady-state (non-inflammatory) conditions. Our study shows that peritoneal MΦs are important producers of many different proteins, including M-lysozyme, apolipoprotein E, fibronectin, serum amyloids, chemokines, TGFβ2, and complement and coagulation components. Interestingly, by comparing the MΦ transcriptome with that of liver, we can conclude that MΦs appear to be the primary producer of many complement and coagulation components. This indicates an intricate interplay among different organs in the regulation of complement and coagulation cascades and suggests that compounds produced locally by tissue MΦs may have an important role in regulating blood coagulation and complement activation. A comparison of the peritoneal MΦ transcriptome with that of human peripheral blood monocytes revealed that these populations show many similarities but also major differences. Altogether, this study is a first step in an attempt to study phenotypic and functional heterogeneity among mammalian monocyte/macrophage subpopulations by quantitative measurements.

## 2. Results

### 2.1. Preparation of RNA from Mouse Tissues, Purified Peritoneal Cell Fractions, and Purified Human Peripheral Blood Monocytes

In the peritoneum, MΦs and B cells represent the main cell populations, constituting approximately 30–40% and 40%, respectively, of the entire peritoneal cell population [20]. The third most abundant cell population is mast cells, which constitute approximately 1–2% of the peritoneal cells. Low numbers of other immune cells including neutrophils and eosinophils can also be detected. To obtain a quantitative estimate of the total transcriptome of MΦs we purified MΦ from peritoneal lavage of thirty mice. For comparison, we additionally purified B cells from the peritoneum, and we also included an analysis of purified mast cells from the same source. Using thirty mice from the same inbred strain, thereby almost genetically identical mice, of the same age and the same living conditions, we reduced the influence on the result by variations between individuals.

MΦs and B cells were purified by FACS using a panel of monoclonal antibodies. Cells were first gated based on forward and side scattering, followed by gating based on positivity for CD19, CD11b, and F4/80 (Figure 1). By using this strategy, well-separated MΦ (large CD11b high and F4/80 high) and B cell (CD19^+^) populations were obtained (Figure 1) [20], followed by preparation of total RNA from the respective populations. RNA was also prepared from a number of mouse Balb/c organs to be used as reference material to evaluate the tissue specificity of MΦ-expressed genes. RNA preparations from the peritoneal MΦs, B cells, ears and lung RNA were then subjected to transcriptome analysis using RNA-seq methodology. Reads were normalized towards the length of the individual mRNAs and listed as a fraction of the entire transcriptome. The same samples were analyzed by the PCR-based mouse Ampliseq transcriptome analysis platform. For the Ampliseq analysis, we included RNA from eight additional tissues as reference samples: brain, tongue, liver, duodenum, pancreas, colon, kidney, and uterus.

As a reference sample representing bone-marrow-derived cells of the monocyte/macrophage lineage, we purified human peripheral blood monocytes (Figure 2), followed by RNA isolation and analysis using the Ampliseq technology. In Table 1, we present the result from one individual. However, we have data from five different individuals of different age and sex that are presented in Appendix A. As can be seen from this Appendix A, the results between individuals are consistent, despite some variations in absolute levels between individuals. However, one remarkable finding was the almost total absence of HLA-B and/or -C expression in some individuals, indicating that some individuals may have a reduced MHC class I repertoire, which may affect their sensitivity to infection by intracellular parasites (Appendix A). We also observed relatively large differences between individuals in the expression levels of the different immunoglobulin Fc receptors (Appendix A).

### 2.2. Analysis of Transcript Levels in Mouse Peritoneal MΦs and Human Peripheral Blood Monocytes

In Table 1, the expression of genes essential for MΦ function are displayed, and a comparison between expression levels in the peritoneal MΦs vs. human monocytes is also included. Further, a comparison of the results obtained using the two different transcriptome approaches (RNA-seq vs. Ampliseq) are displayed. We focused on the following categories of genes: the most highly expressed genes, antibacterial proteins, receptors, cell adhesion molecules, cytokines, chemokines and cytokine/chemokine receptors, complement factors, coagulation factor cell signaling molecules, and transcription factors to obtain a detailed picture of the biological function of this population of mouse peritoneal MΦs. This analysis revealed the most abundant transcripts of the MΦs are those coding for lysozyme (an antibacterial protein), fibronectin (a connective tissue component), serum amyloids (i.e., Saa3 and Saa1), apolipoprotein E, arachinodate15-lipoxygenase (Alox15), and a number of different lysosomal proteases (i.e., cathepsins B, D, L, A, S, Z, and H) (Table 1, (A, B, D, I)). High levels of additional lysosomal proteins were also observed, including lysosomal membrane protein 5 (Laptm5), prosaposin (Psap) and alpha-mannosidase (Man2b1) (Table 1, (F)). Relatively high levels of lipid mediator-related enzymes were also observed, including phospholipase A2 (Pla2g7), prostaglandin I synthase (Ptgis) and arachinodate-5-lipoxygenase (Alox5) (Table 1, (B)). 

By using this strategy, we noted some apparent discrepancies between the two transcriptome approaches. Hence, the RNA-seq approach appears suitable for distinguishing closely related genes, such as M and P lysozymes (Lyz1 and Lyz2), whereas the Ampliseq method was able to detect genes having high sequence divergence as exemplified by the MHC class I alpha chain and the MHC class II alpha and beta chains. As judged by the RNA-seq data, the peritoneal MΦs appear to only express the M-lysozyme (Lyz1), whereas the Ampliseq method appears to be less capable of differentiating between Lyz1 and 2 (Table 1, (A)). Further, the RNA-seq approach indicated expression of both serum amyloid 3 and 1 (Saa3 and Saa1), whereas Ampliseq detected Saa3 only (Table 1, (A)). It was also noted that the Ampliseq method appeared to be superior to the RNA-seq approach for detecting the MHC class I alpha chain and the MHC Class II alpha/beta chains, again suggesting that the Ampliseq method is well suited to detect highly variable molecules (Table 1). 

In contrast to the slightly disparate results for selected genes (see above), both techniques provided highly similar results for the absolute majority of genes analyzed including several cell adhesion molecules such as integrin alpha-m (Itgam), integrin alpha-6 (Itga6), integrin beta-2 (Itgb2), P-selectin (Selp), and ICAM2 (Table 1, (P)). Both approaches also reveal high levels of the MHC class I component beta-2 microglobulin, FcgRIII, the signaling component of the Fc receptors (i.e., Fcer1g), and several protease inhibitors including cystatin C, SLPI, Serpin B2, and Timp2. We also observed high levels of FcRN, the transport receptor for IgG but low levels of Fc-gamma receptor 4 (Fcgr4) and even lower expression of the high affinity receptor for IgG (Fcgr1) (Table 1, (J)). Contradicting results were obtained for the negatively regulating Fc receptor for IgG (Fcgr2b), for which we noted a high expression level based on RNA-seq but only low levels based on Ampliseq analysis (Table 1, (J)).

Among the most highly expressed transcripts, we also found several key components of the complement and coagulation systems. High expression of genes encoding the complement component C1q (C1qA, C1qB, and C1qC) was found based on the RNA-seq data, whereas the Ampliseq approach revealed primarily expression of C1q-B (Table 1, (E)). High levels of C4a, C4b, properdin (Cfp), and factor H were also seen (Table 1, (E)). 

We also noted high expression of cytochrome b245 (Nox2/cytochrome b558), a compound with a role in the formation of reactive oxygen species as part of the antibacterial and antiviral responses mediated by MΦs (Table 1, (A)). Among the highly expressed transcripts, we also found PAD-4 (Padi4). PAD-4 is essential for the deamination of arginine on histone H3, converting it to citrulline, thereby reducing the charge of the histones. This process is essential for formation of neutrophil extracellular traps (NETs), a process where decondensed chromatin is expelled into the extracellular environment together with granule protein. After their initial discovery in neutrophils, formation of extracellular DNA traps has also been reported in several other innate immune cells (Table 1, (C)) [21]. High levels of transcripts for several proteins thought to be involved in phagocytosis were also observed, including filamin B and A, CD209b, and Timd4, the latter a receptor for phosphatidyl serine exposed on the surface of apoptotic cells (Table 1, (C)). However, we neither detected transcripts for the defensin family of antibacterial peptides or for cathelicidin nor did we detect transcripts for neutrophil granule proteins (i.e., myeloperoxidase, N-elastase, proteinase 3, and cathepsin G) (Table 1, (C) and data not shown).

Of the scavenger receptors, only very low levels of Marco, CD177, CD163, Scarb1, and Scara 3 and 5 were observed, whereas higher expression of Scarb2, CD36 (Scrb3), and CD68 was seen (Table 1, (M)). Of the classical monocyte/MΦ surface markers, we found relatively high levels of CD14, lower levels of CD40 and CD86 (B7-2), and even lower levels of CD80 (B7-1) (Table 1, (L)). As expected, expression of the T-cell marker CD28 and of the dendritic cell activation marker, CD83, was essentially undetectable (Table 1, (L)).

By comparing the expression levels of this panel of mouse MΦ transcripts with human peripheral blood monocytes, we observed a marked difference between these two cell populations in many aspects but also clear similarities. Notably, lysozyme was the most highly expressed transcript in both cell types. Similar to the MΦs, we found high expression of filamin A, cytochrome b245, and serglycin (Table 1, (B and C) and Appendix A) in the monocytes. However, in contrast to the MΦs, the monocytes did not express significant amounts of amyloids (Table 1, (A)), and completely lacked the expression of fibronectin, lubricin, syndecan 3, and extracellular matrix protein 1 (Table 1, (B)). 

There was also a major difference concerning the lipid mediators (Table 1, (D)). In contrast to the mouse peritoneal MΦs, the monocytes expressed only low levels of phospholipase A2, Alox15, prostaglandin I synthase (Ptgis), and apolipoprotein E (ApoE) (Table 1, (D)). On the other hand, similar to the peritoneal MΦs, no expression of defensin or cathelicidin was detected in the monocytes (Table 1, (C)). When examining the expression of complement and coagulation components, we observed major differences between the two cell populations. Whereas both populations express high levels of properdin and ficolin, the monocytes, in contrast to the MΦs, did not express C1q, C4a, C4b, or complement factor H, and they also lacked expression of all coagulation components (Table 1, (E and F)). Both populations expressed high levels of the lysosomal proteins Laptm5, Psap, and cathepsins B, D, and S, whereas the monocytes showed low levels of cathepsins L and A and essentially lacked expression of the alpha-mannosidase (Ma2b1) (Table 1, (I)). 

The two populations showed similarities in the expression of immunoglobulin Fc receptors. Mouse peritoneal MΦs expressed high levels of FcgRIII, whereas human monocytes had a high expression of FcgRIIa. Both populations expressed low levels of FcgRI and high levels of the signaling component FceRIg (Table 1, (J)). Further, it is noteworthy that both expressed high levels of FcRN (Table 1, (J)) as well as relatively high levels of MHC Class I molecules but differed markedly in their expression of MHC Class II molecules (Table 1, (K)): the monocytes showed high levels of both HLA-DR and HLA-DP but almost completely lacked HLA-DQ, whereas the mouse MΦs expressed low levels of MHC Class II (Table 1, (K)). Similar to the mouse MΦs, the monocytes expressed high levels of CD14 (Table 1, (L)) and relatively high levels of B7-2 but low levels of B7-1 (Table 1, (L)). The monocytes also expressed low levels of Relma (Retln; a receptor thought to regulate TH2 immunity) compared to the mouse MΦs (Table 1, (L)) [22]. Concerning scavenger receptors, both populations showed low levels of Marco and CD177 and different Scara members but high levels of CD68 and CD36 (Table 1, (M)). The human monocytes expressed higher levels of CD163 compared to the mouse peritoneal MΦs (Table 1, (M)). With regard to cytokine, chemokine, and endothelin receptors, some major differences were noted. Mouse MΦs expressed high levels of the FGF receptor 1, whereas the monocytes almost completely lacked expression of this receptor (Table 1, (N)). Further, the monocytes expressed low levels of the M-CSF receptor compared to the mouse MΦs (Table 1, (N)). The monocytes also showed very low levels of GM-CSF receptor expression compared to the MΦs (Table 1, (N)). In contrast, whereas monocytes showed a high level of G-CSF receptor expression, the peritoneal MΦs expressed low levels (Table 1, (N)). Both populations showed high levels of Tnfrsf1b (a TNF receptor) (Table 1, (N)). Both populations also show relatively high levels of IL-10 receptor expression (Table 1, (N)). Relatively low levels of expression were seen for most of the other cytokine receptors in both MΦs and monocytes (Table 1, (N)). 

When analyzing the expression of cell adhesion molecules, we also noted several major differences between the two populations. Of these, the top transcript in MΦs was integrin alpha m with considerably lower expression in the monocytes (Table 1, (P)). Low levels of integrin beta1 expression was also seen in the monocytes vs. the MΦs (Table 1, (P)). A similar situation was seen for P-selectin, Emilin2, and ICAM-2 with high levels in MΦs and lower expression in monocytes (Table 1, (P)). With regard to chemokines and cytokines, we noted major differences. Platelet factor 4 was expressed at very high levels in the peritoneal MΦs but at low levels in the monocytes (Table 1, (Q)). Ccl6, Ccl9 Cxcl13, and Cxcl14 were all expressed at high levels in the mouse MΦs but were almost totally absent in the monocytes (Table 1, (Q)). The only cytokines/chemokines for which significant expression was seen in the monocytes were Cxcl16, Tgfb1, TNF-alpha, and Vegfa (Table 1, (Q)). 

Of selected signaling components, Tyro binding protein (Tyrobp) was expressed at significant levels in the monocytes (Table 1, (R)). Among selected transcription factors, we found significant expression levels for Pu.1, Runx3, and Zab2. Notably, none of the GATA factors were expressed at significant levels (Table 1, (R)). In the peritoneal MΦs, only GATA6 was expressed at significant levels (Table 1, (S)). 

### 2.3. Analysis of Transcript Levels for a Panel of Pattern Recognition Receptors and Proteins Involved in Angiogenesis

The expression of receptors involved in sensing microbial components, including Toll-like receptors, NOD, and Rig receptors as well as several transcripts involved in angiogenesis, is summarized in Table 2. For comparison, we included data from mouse peritoneal mast cells, B cells, and human blood monocytes in this analysis. As seen in Table 2, the expression levels for all these transcripts are remarkably low in all four analyzed cell populations, in agreement with previous studies of mast cells [23]. For some of these microbial sensors we see somewhat higher levels of transcripts, including TLR-4, -13 -7, and -8 (Table 2, (A)). MDA5 and Rig 1 were both expressed in all four cell populations but at relatively low levels (Table 2, (A)). Angiogenesis-related transcripts were low to undetectable in the MΦs, suggesting that the peritoneal MΦs may not have an important role in angiogenesis (Table 2, (B)). In contrast, mast cells express relatively high levels of such transcripts, including Vegfa and Vegfb. Mast cells but neither MΦs nor B cells also express significant levels of angiopoietin 1 but not angiopoietin 2 (Table 2, (B)). Notably, different to the peritoneal MΦs, the monocytes express Vegfa (Table 2). 

### 2.4. Analysis of Transcript Levels in the Mouse Liver

As detailed above, a number of plasma components, including antibacterial proteins, amyloids, and complement components, were identified in the MΦ transcriptome. The liver is generally implicated as the main producer of the majority of such plasma proteins, and it was therefore of interest to compare the expression levels of such compounds between the liver and the peritoneal MΦs. To this end, we also performed a corresponding transcriptome analysis of the liver and of a number of other organs as reference material. 

The liver consists of a number of cells, including hepatocytes, liver MΦs (named Kupffer cells), liver endothelial cells (LECs), and fat cells. Hepatocytes are the major population of the liver, and most of the transcripts in the liver thereby originate from this cell population. Notably, no or very low expression of the serum amyloids 3 and 1, C1qA, C1qB, C1qC, properdin or ficolin was observed in the liver transcriptome. The main amyloid of the liver was instead Saa4 but was expressed at lower levels compared with the expression of Saa3 and 1 by the MΦs (Table 3, (A) and Table 1, (A)). Instead, our transcriptome analysis indicated that the liver is the main producer of the majority of both complement and coagulation factors, including fibrinogen (Fgb, Fga and Fgg), thrombin (F2), and coagulation factors V, VII, X, XII, and XIII. Low levels of factors IX, XI, and VIII were also observed (Table 3B). The most highly expressed transcripts for complement proteins were factor 3 (C3), factor H (Cfh), factor 4 (C4b), C4-binding protein (C4bp), factor 8 gamma (C8g) factor 5 (Hc), C-reactive protein (Crp), C1r, and the C1q-binding protein (C1qbp) (Table 3, (C)). Lower levels of factor 8b and factor 8a were seen, along with very low levels of C1qb, C2, ficolin (Fcna), and properdin (Cfp) (Table 3, (E)). The latter most likely originate from Kupffer cells of the liver, as these components are expressed at very high levels by the peritoneal MΦs (Table 1). Coagulation factor VIII is most likely almost exclusively produced by the liver endothelial cells (LECs) [27]. This coagulation factor is also produced at low levels by the kidneys and in the uterus (Table 3, (D)). As expected, liver is the major producer of albumin (Table 3, (A)). Very high expression levels were seen for different enzymes involved in lipid biosynthesis, including stearoyl coenzyme A desaturase (Scd); the fatty-acid-binding protein (Fabp1); the apolipoproteins A1, C1, C3, and E; the retinol-binding protein (Rbp4) (Table 3, (B)). Genes coding for enzymes involved in amino acid metabolism were also highly expressed (Table 3B). 

### 2.5. Analysis of Transcripts Representing a Panel of Signature Genes Identified by Single Cell Analysis of Monocytes, Dendritic Cells, and MΦs from Different Tissues

A number of signature genes have been identified by single cell analysis of monocytes, dendritic cells, and different MΦ subpopulations originating from different mouse tissues [19]. In Figure 3, we placed our data in the context of these earlier studies. As can be seen, both types of studies matched remarkably well; all except five of the transcripts previously identified as signature genes for peritoneal MΦs by single cell analysis were significantly expressed in the peritoneal MΦs in the present study. In contrast, relatively few of the signature genes for other MΦ subpopulations, dendritic cells, and monocytes were significantly expressed based on our analysis of peritoneal MΦs (Figure 3).

Based on the expression of CD11b (Itgam) and F4/80 (Adgre1) in the mouse peritoneal MΦs, we can confirm that the MΦ population we analyzed was the large CD11b high and F4/80 high population of mouse peritoneal MΦs (11,106 and 1168 reads, respectively, for CD11b and F4/80 (Table 1, (P and L)) [20].

## 3. Discussion

Analysis of the total transcriptome by several recently developed platforms have made it possible to, with high resolution, analyze the expression levels of all the genes within the entire genome of a species. However, there were some difficulties involved in the way such analyses were performed. In addition, in the majority of the published transcriptome analyses, the information was generally presented in the form of heat maps, which provide relative values compared to other tissues, sample conditions, etc. Transcripts differing by several orders of magnitude in expression levels are in these heat maps often depicted in the same color, bright red or bright blue. By this type of presentation, valuable quantitative information is lost.

During this and previous work, we experienced a high impact of the reference library used for the RNA-seq analysis when generating the final data used for comparative studies. Genomic reference libraries have a large tendency to give high error frequency and should, based on our experience, therefore be avoided. Instead, good global mRNA transcriptomic libraries provide the most reliable data for these types of studies.

In a previous study, we attempted to validate transcriptome data by applying different methods. In a study of the mouse mast cell transcriptome, we compared three different independent strategies, and we were able to show that they gave highly similar results [23]. Hence, based on such a systematic comparison of independent methods for transcriptomic analyses, we are confident that the data presented here represent reliable quantitative information concerning MΦ-related transcripts. However, even after performing such a validation, there are potential pitfalls. For example, in our RNA-seq data, the highly variable molecules MHC Class I and II were not detected, most likely due to the fact of that their high variability results in difficulties of detection. The sequence reads may not match 100% to the corresponding sequence in the reference library and were therefore not counted. In contrast, these genes were detected by the Ampliseq approach, and it thus appears that the Ampliseq methodology was less sensitive to minor sequence differences. However, the Ampliseq method appears less reliable for distinguishing between highly homologous transcripts, e.g., between Lyz1 and Lyz2, between C4a and C4b, and between C1qA, C1qB, and C1qC. However, by combining the two technologies we can overcome most of these problems and obtain reliable data concerning the transcriptome of the studied tissues and cells.

To obtain a sufficient number of cells to obtain a good coverage of the transcriptome and thereby high-quality quantitative information, we collected peritoneal cells from thirty mice. Using thirty mice from the same in-bred strain, and thereby almost genetically identical mice of the same age and the same living conditions, we also reduced the influence on the results by variations between individuals and obtained a good estimate of the transcription levels of all the transcripts within these cell populations in this strain of mice.

Based on the results from the two transcriptome platforms, we noted that these large CD11b high and F4/80 high peritoneal MΦs are potent producers of a number of different proteins, including lysozyme, Saa3, and Saa1, whereas they almost completely lack Saa4 (Table 1, (A)). Saa4 is instead the major serum amyloid produced by the liver (Table 3). The human monocytes seemed to lack expression of all of the amyloids (Table 1, (A) and Appendix A)). Based on the RNA-seq data we see that, as expected, MΦs exclusively express the M-Lysozyme (Lyz1).

MΦs are highly mobile cells and also active phagocytes. Molecules that are important for both of these processes were expressed at very high levels, including cell adhesion molecules such as the integrins alpha-M, alpha-6, beta-2 and beta-1, P-selectin, and ICAM-2 (Table 1). MΦs are also the major phagocytic cells of the body, and in this capacity, the lysosomes play a central role. In line with this, we found high expression of several lysosomal proteases (cathepsins B, D, L, A, S, Z, and H) in the peritoneal MΦs (Table 1, (I)). It was also interesting that the peritoneal MΦs expressed high levels of a number of complement components and also a few components of the coagulation system. In contrast, the corresponding transcripts were almost totally absent in the liver transcriptome, indicating that the peritoneal MΦs are the prime producers of several of these components. This indicates a complex pattern of regulation of both of these systems, including the production both centrally by the liver and locally in the various tissues by MΦs.

We also observed very high levels of transcripts for fibronectin in the peritoneal MΦs (Table 1). This is in agreement with earlier studies at the protein level [28]. Generally, it is thought that liver is the major source of plasma fibronectin and that fibroblasts are the main producers of tissue fibronectin. However, based on the present study, peritoneal MΦs express ~25 times higher levels of fibronectin compared with liver. Hence, peritoneal MΦs are probably important local producers of this protein. We also found that MΦs produce high levels of additional connective tissue components, including lubricin (Prg4), syndecan-3 (Sdc3), and extracellular matrix protein 1 (Ecm1) (Table 1, (B)). All of these were essentially absent in monocytes, indicating a major difference between blood monocytes and tissue MΦs with regard to connective tissue homeostasis (Table 1, (B)). We also noted that PAD-4 (of major importance for the formation of NETs) was expressed at much higher levels in MΦs vs. monocytes, indicating differences in the ability of the respective populations to produce DNA containing extracellular traps.

MΦs in general proliferate in response to M-CSF. In line with this, the peritoneal MΦs were found to express high levels of the M-CSF and the GM-CSF receptors but relatively low levels of G-CSF receptor (Table 1, (N)). Interestingly, and unexpectedly, we observed the opposite for the human monocytes, where the G-CSF receptor was the dominating cytokine receptor (Table 1, (N) and Appendix A). The significance of this is not known but indicates a marked difference in cytokine regulation between the two populations.

MΦs are found in all tissues of the body, where they are adopted for particular functions related to the respective tissue. A major question in the field of MΦ biology has been to outline differences in phenotype and function between these MΦ subpopulations. By single cell analysis, clear differences were identified between MΦ subpopulations. For example, intestinal MΦs differ substantially in phenotype from all of the other MΦ subpopulations, and dendritic cells appeared as a separate cluster distinct from both the majority of yolk-sac-derived MΦ subpopulations and intestinal MΦs [19]. Gene expression patterns that specify these different subpopulations have recently been revealed [19]. 

To obtain insight into possible differences between these MΦ populations, we specifically looked at all the transcripts listed for these different mouse MΦ, monocyte, and DC subpopulations and listed the expression levels of these marker genes in a table for more easy comparison. All the transcripts with a level higher than 50 reads were marked orange in Figure 3. As can be seen from the Figure 3, our data confirm most of the data from the single cell analysis. Almost all of the marker genes identified by single cell analysis for mouse peritoneal MΦs were expressed at higher than 50 reads in our analysis (Figure 3). However, there were five genes that did not seem to fit the single cell data, i.e., Car6, Cyp26a, Lbp, Rarb, and Sox7, as all of them showed relatively low levels of expression in our analysis of the peritoneal MΦs (Figure 3). A few of the marker genes for other populations were also expressed at a level of more than 50 reads in our analysis of the peritoneal MΦ population. There were five such genes in the lung marker gene list, 11 in the Kupffer cell list, seven in the monocyte list, 10 in the microglia list, none in the intestinal MΦ list, and only one in the list of the DCs (Figure 3). However, when we look at these marker genes a majority are actually expressed at relatively high levels in most tissues analyzed, indicating that they are poor representatives as marker genes.

An interesting finding from the single cell studies of MΦs was that the transcriptomes of various MΦ subpopulations differed substantially, indicating major functional differences. From such studies, it was evident that the MΦ population that was the most different from the peritoneal MΦs was the intestinal MΦs and the dendritic cells and, to a slightly lesser extent, the monocytes (Figure 3). To obtain more direct evidence for similarities and differences between these different MΦ populations, an in-depth quantitative analysis of expression levels of the entire transcriptome of these subpopulations would be very informative to more specifically identify the major differences in biological function between these populations. It has for example been shown that the two major subpopulations of MΦs in the peritoneum, the large CD11b high and F4/80 high, analyzed in this communication, and the small CD11b high and F4/80 low populations had, at least partly, different biological functions [20,29]. The large peritoneal MΦs seemed to originate from the yolk sac and were self-renewing, whereas the minor population of small F4/80 low population seemed to be monocyte derived and better at antigen presentation to naive T cells compared to the major large F4/80 high subpopulation [30]. In line with this finding the large peritoneal MΦs showed a low level of expression of MHC Class II as is also shown here, whereas the small peritoneal MΦs expressed relatively high levels of MHC Class II [20,31]. The large peritoneal MΦs seems instead to be the most important cell type, of these two, in the clearance of bacterial cells during abdominal sepsis [31]. However, they seem also to be a potential reservoir of *Staphylococcus aureus*, as these bacteria seem to survive inside the MΦs. By removing the bacteria from the peritoneal cavity by phagocytosis, they delay the influx of neutrophils, which seems to be essential for the clearance of the bacterial infection [32].

We also observed some major differences in the expression of transcription factors. The only GATA factor expressed by the mouse peritoneal MΦs was GATA-6, whereas human blood monocytes were negative for all GATA factors. The peritoneal MΦs also expressed Pu.1 and Zeb2 at high levels and lower levels of Runx1 (Table 1, (S)). The major transcription factors expressed by the human monocytes were also Pu.1 and Zeb2, and monocytes also expressed Runx3 (Table 1, (S)). Other transcription factors are most likely involved, but the ones highlighted here are those that most clearly separated the MΦs from the other mouse cells and tissues included in the analysis. Several of these transcription factors have been shown to control the expression of the M-CSF receptor, a receptor with a major role in the proliferation, differentiation, and survival of cells of the MΦ lineage [33].

Concerning Fc-receptors, we also obtained several interesting new findings. It is known that FcgRIII is expressed by mouse MΦs and that FcgrIIA is highly expressed by human MΦs and monocytes, and it is also known that low levels of the high affinity Fc receptor FcgRI can be found on both populations. However, the presence of FcRN on either of the populations has, to our knowledge, not previously been reported, but further work is required to determine the function of this receptor in a MΦ/monocyte context. The expression levels of these receptors differed also quite extensively between individuals in the human monocytes, indicating that they may vary depending on inflammatory status of a person, even if these donors were all healthy blood donors (Appendix A).

Interesting was also the very low levels in all analyzed cell populations of the pattern recognition receptors: the RIG, NOD, and Toll-like receptors. These low levels were apparently still sufficient for a rapid response by these cell populations and the question is if they remain at the same levels also after response to, for example, LPS interaction with TLR-4 or if there is a strong up- or downregulation of these receptors upon responding to the ligand.

In summary, we here presented a detailed quantitative map of one population of mouse MΦs, the large peritoneal MΦs, and placed this in the context of the transcriptome of human peripheral blood monocytes and the total mouse liver transcriptome. We analyzed approximately 240 different transcripts and obtained quantitative measurements of their expression levels in these three tissues and also made comparisons with eight other tissues and of mouse peritoneal B cells. This information can now serve as a roadmap to study phenotypic and functional differences between different subpopulations of cells of the monocyte/macrophage lineage. The most interesting finding was the apparent major role of tissue MΦs in both the complement and coagulation systems and the major difference between monocytes and MΦs concerning their role in connective tissue homeostasis. We also observed a major difference in the steady-state levels of MHC Class II, a molecule central for antigen presentation. Monocytes seem here to be ready to perform this task without prior activation, whereas tissue MΦs needs activation to become active antigen-presenting cells. Although some of the differences observed may depend on species-specific differences, we can conclude that there are major differences in the transcriptome and thereby also the in vivo function of tissue MΦs and circulating monocytes.

## 4. Materials and Methods

### 4.1. Mice

Female BALB/c mice were purchased from Taconic Biosciences (Ejby, Denmark) and maintained at the animal facility of the Biomedical Center (Uppsala University). The animal experiments were approved by the local ethics committee (Uppsala djurförsöksetiska nämnd; Dnr 5.8.18-05357/2018).

### 4.2. Peritoneal Cell Extraction and FACS Sorting of Peritoneal Macrophages and B Cells

For the extraction of peritoneal cells, thirty mice were euthanized by neck dislocation during isoflurane anesthesia, the abdominal skin was removed, and 9 mL of ice-cold phosphate-buffered saline (PBS) was injected into the peritoneal cavity. After making sure that the injected PBS was thoroughly dispersed within the peritoneal cavity, peritoneal lavage fluid was collected, and the cells were pelleted by centrifugation at 400× *g* for 10 min. The cells were resuspended in PBS (pH 7.4) with 2% heat-inactivated fetal bovine serum (Gibco, Carlsbad, CA, USA), followed by incubation with the following fluorescent-labeled antibodies: F4/80 (BM8), CD11b (M1/70), CD19 (1D3), CD117 (2B8), and FcεRI (MAR-1). The antibodies were obtained from BD Biosciences (Franklin Lakes, NJ, USA) or eBioscience (Hatfield, UK). FACS-isolated peritoneal macrophages and B cells were collected for RNA isolation. The flow cytometry-based cell sorting was performed on a FACSAria III (BD Biosciences), and data were analyzed with FlowJo software (TreeStar Inc., Ashland, OR, USA).

### 4.3. Isolation of Human Peripheral Blood Monocytes by Magnetic Cell Sorting

Peripheral blood monocytes were isolated from peripheral blood obtained from healthy donors at the Akademiska Hospital in Uppsala, Sweden, in the form of buffy coats. PBMCs were isolated using Ficoll–Paque Plus (GE Healthcare, Uppsala, Sweden) and standard density gradient centrifugation. PBMCs were further washed with PBS containing 2 mM of EDTA and incubated with anti-CD14-coated magnetic beads (Miltenyi Biotec, Bergisch Gladbach, Germany). Positive selection of CD14^+^ cells was performed through magnetic cell separation. Subsequently, CD14 cells were stained with anti-human CD14 PE antibody (clone: 61D3, Invitrogen, Carlsbad, CA, USA), and the purity was verified (over 90%) on a MACSQuant VYB Flow Cytometer (Miltenyi Biotec). Approximately 4 million of these cells were immediately pelleted, and the total RNA was purified by a standard protocol.

### 4.4. RNA Isolation

Total RNA was prepared from FACS-sorted cells and CD14^+^ monocytes using the Nucleospin RNA kit from (Macherey-Nagel, Duren, Germany), according to the manufacturer’s recommendations. The RNA was eluted with 30 μL of DEPC-treated water, and the concentration of RNA was determined by using a NanoDrop ND-1000 (NanoDrop Technologies, Wilmington, DE, USA). Later the integrity of the RNA was confirmed by visualization on 1.2% agarose gel using ethidium bromide staining.

Ear, lung, liver, brain, heart, tongue, pancreas, duodenum, colon, kidney, uterus, and spleen tissues were carefully dissected from the mouse. Immediately after removal from the animal, the tissues were frozen in liquid nitrogen and ground into a fine powder with a pestle in a mortar. The tissue powder was then used for total RNA isolation using the same protocol as for the cell fractions described above.

### 4.5. Analysis of the Transcriptome by RNA-seq and by the Thermo Fisher Ampliseq Chip and PCR-Based Method

Total RNA from the different cell fractions and whole tissues were sent to GATC-Biotech (Konstanz, Germany) for transcriptome analysis. The procedure was that they purified mRNA by poly A selection following fragmenting of the RNA and then performing sequencing of 20–30 million fragments. The individual reads of a length of in general 50–100 nucleotides are then matched against a reference library. After testing several strategies and reference libraries, the results from the sequencing were run against a transcriptome reference. This transcriptome reference resulted in highly reliable data, which matched well with previous cDNA library screenings and later also with the Thermo Fisher chip-based Ampliseq transcriptomic platform at the SciLife Lab in Uppsala, Sweden. The number of reads per gene was, for the RNA-seq data from GATC, then adjusted to the transcript length as longer transcripts generate more fragments per mRNA and, thereby, a higher number of reads. The Thermo Fisher Mouse Ampliseq transcriptome analysis platform is based on the purification on a chip of the individual mRNAs, which are then PCR amplified and sequenced individually. The RNA is not fragmented, which is why, in general, every mRNA was read only once and the number of reads then matched the expression level more directly.

## Figures and Tables

**Figure 1 ijms-23-01185-f001:**
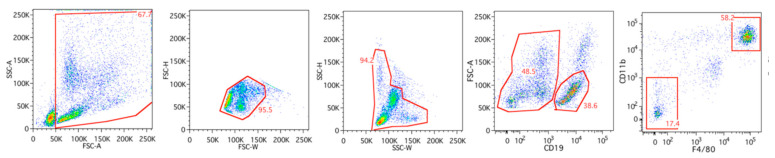
Gating strategy used for identification and sorting of peritoneal macrophages (i.e., CD19^−^, CD11b^+^, and F4/80^+^) and B cells (CD19^+^ and low FSC-A).

**Figure 2 ijms-23-01185-f002:**
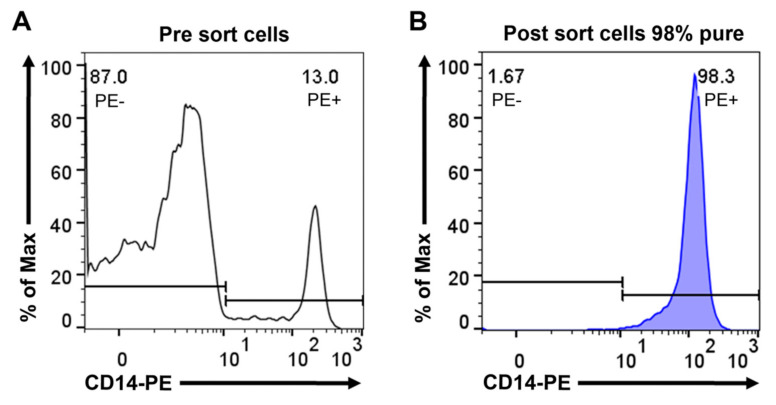
Purity of human peripheral blood monocytes obtained from PBMCs by magnetic cell sorting using CD14 microbeads. Separated cells were stained with anti-human CD14 PE antibody and analyzed by flow cytometry. Representative flow cytometry histograms show PBMCs before sorting (**A**) and cells after sorting (**B**).

**Figure 3 ijms-23-01185-f003:**
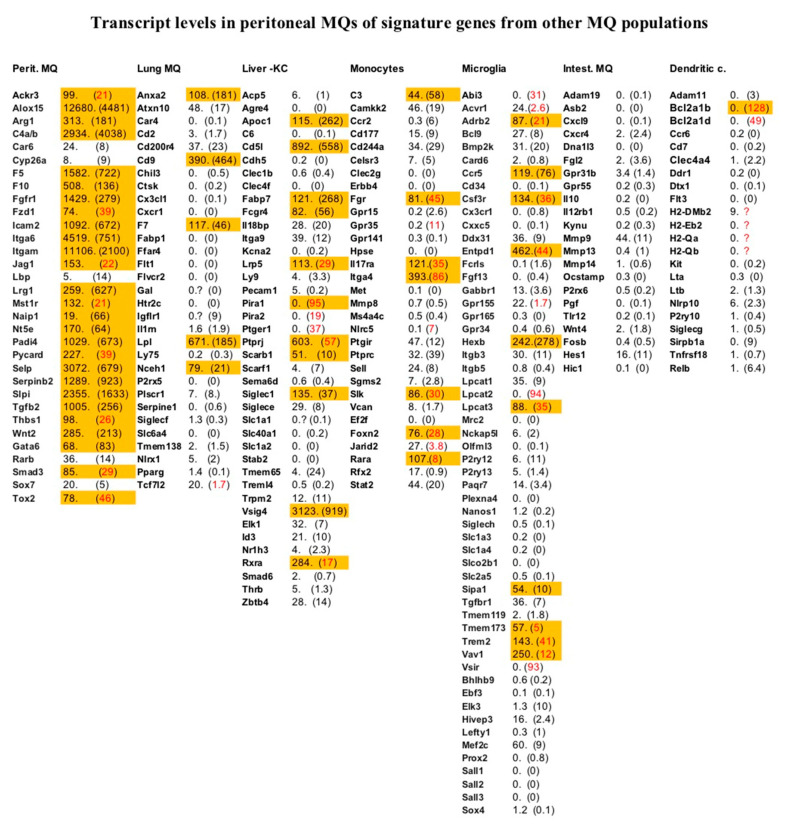
Signature gene sets identified for different MΦ subpopulations by previous single cell analysis. Transcript levels in the peritoneal MQs for a number of genes previously identified as signature transcripts for a few different MQ, monocyte, and dendritic cell populations by single cell analysis as summarized by Summers et al. [19]. The number of reads for each of the different genes are given in actual numbers we obtained from the Thermo Fisher Ampliseq analysis. The numbers in brackets are the numbers obtained from the GATC RNA-seq analysis. When the number or reads in one of these two studies are higher than 50 the transcript is marked in orange. When there is a major difference between the two analysis methods the RNA-seq value is marked by red text.

**Table 1 ijms-23-01185-t001:** Transcript levels for genes expressed in mouse peritoneal MΦs. The analysis highlights genes that are highly expressed in MΦs, genes that are selectively expressed in MΦs, and genes of particular biological relevance for MΦ function. The number of reads for each of the different proteins are given in actual numbers obtained from RNA-seq and Ampliseq analyses. In the RNA-seq analysis, the same transcript occasionally appeared several times due to the existence of splice variants. In these cases, the sum of the differential read values are presented within brackets. Genes for which we saw low or no expression in the MΦs, including defensins, cathelicidin, histidine-rich glycoprotein, histidine decarboxylase, and VEGFs, were added to the list as reference material. As a reference sample, we also included Ampliseq data for the same molecules from MACS-purified (anti-CD14) human peripheral blood monocytes. In cases where there were species-specific transcripts, mouse genes not found in the human genome or human genes not found in the mouse genome, these are marked with a short line.

	Mouse Peritoneal MΦs		H-Monocytes
	RNA-Seq	Ampliseq	Ampliseq
** A. Amyloids and General Transcripts **			
Saa3 (serum amyloid, apolipoprotein)	10,137	7412	0
Saa1 (serum amyloid, apolipoprotein)	9084	0 *	0
Saa2 (serum amyloid, apolipoprotein)	0	4	0
Saa4 (serum amyloid, apolipoprotein)	16	0	0
Actb (beta actin)	3425	7060	19,693
Tcn2 (Transcobalmin)	(1027)	5383	21
Wfdc17 (WAP protein domain protein-activated MQ)	4897	4064	0
Tln1 (Talin 1 cytoskeletal membrane connector)	370	1971	248
Itsn1 (Intersectin 1 membrane trafficking)	(276)	1359	0
Grn (Granulin)	1817	2744	588
Bst1 (ADP-ribosyl cyclase 2)	159	1204	100
Gda (Guanine deaminase)	523	1232	0
Hamp (Hepsidin Iron import)	148	876	1
Ninj1 (Ninjurin 1 apoptosis signal?)	390	1122	578
Hal (Histidine ammonia lyase)	335	1561	0
Hdc (Histidine decarboxylase)	21	42	0
Hrg (Histidine rich glycoprotein)	0	0	0
** B. Extracellular Matrix **			
Fn1 (Fibronectin)	(10,119)	25,920	0
Prg4 (Proteoglycan 4, Lubricin)	3921	3606	0
Srgn (Serglycin-proteoglycan core protein)	1022	2803	3855
Sdc3 (Syndecan 3)	873	3205	5
Ecm1 (Extracellular matrix protein 1)	(1876)	3180	1
** C. Antimicrobial Proteins **			
Lyz1 (M-Lysozyme)	8354	791	27,394
Lyz2 (P-Lysozyme)	0	104,081 *	-
Defb (Beta-defensins)	0	0	0
Camp (Cathelicidin)	0	0	1
Cybb (Cytochr.b-245 (Nox2) Cytb558)	1018	2664	1081
Padi4 (Peptidyl arginine deiminase type IV)	673	1029	77
Flnb (Filamin B, fagocytosis)	(419)	2235	9
Flna (Filamin A)	(287)	1458	1175
Timd4 (Binds Phosphatidyl serine, apoptotic cells)	933	1244	0
** D. Lipid Mediators and Metabolism **			
Alox1 (Arachinodate-15-lipoxygenase)	4481	12,680	0
Pla2g7 (Phosplipase A2)	1325	1549	15
Alox5ap (Arachinodate-5-lipoxygenase activating protein)	(1058)	1296	111
Ptgis (Prosaglandin I syntase)	387	1283	0
Alox5 (Arachinodate-5-lipoxygenase)	178	494	190
Dpep2 (Dipeptidase 2 membrane bound, incl. PGD4)	(360)	1243	77
ApoE (Apolipoprotein E)	(14,181)	2413	2
Pltp (Phospholipid transfer protein)	2036	5788	1
Plin2 (Perilipin2 cytopl. lipid droplet binding)	426	1370	777
Retnla (Resistin like alpha, cholesterol hom?)	609	1223	0
Smpdl3a (Sphingomyelin Phosphodiesterase acid-like 3)	1103	2505	3
Lipn (Lipase important for keratinocytes)	34	11	11
** E. Complement Proteins and Their Receptors **			
Cfp (Complement factor P, Properdin)	2941	5225	991
C1qa (Complement factor C1q A)	3661	4 *	7
C1qb (Complement factor C1q B)	2123	3978	3
C1qc (Complement factor C1q C)	2127	128 *	1
C4b (Complement factor 4B)	3087	2934	0.1
C4a (Complement factor 4A)	951	28 *	1
Cfh (Complement factor H)	739	1980	0
Fcna (Ficolin A) (human Ficolin 1, Fcn1)	1428	1306	3198
Vsig4 (V-Ig domain cont.4 Comp C3b rec)	919	3123	5
C3 (Complement factor 3)	58	44	2
CFB (Complement factor B)	205	0 *	0.2
C2 (Complement factor 2)	4	5	14
C3ar1 (C3a receptor)	139	290	41
C5ar1 (C5a receptor 1)	(104)	68	0
** F. Coagulation Proteins **			
F5 (Coagulation factor V)	722	1582	40
F10 (Coagulation factor X)	(266)	508	0
F7 (Coagulation factor VII)	46	117	0
F12 (Coagulation factor XII)	0	0	1
F9 (Coagulation factor IX)	0.1	0	0
F2 (Thrombin)	0	0	0
** G. Proteases **			
Mmp19 (Matrix metalloprotease 19)	(18)	81	1
Mmp9 (Matrix metalloprotease 9)	11	42	9
Mmp27 (Matrix metalloprotease 27)	6	10	0
Mmp12 (Matrix metalloprotease 12)	3	0.3	0
** H. Protease Inhibitors **			
Cst3 (Cystatin C)	3497	5347	3704
SLPI (Secretory leukocyte protease inhibitor)	1633	2355	0.4
Serpinb2 (Serpin B2)	(915)	1289	38
Timp2 (Metalloproteinase Inhibitor 2)	821	2135	283
Timp1 (Metalloproteinase Inhibitor 1)	0.1	0	725
Serpinb9 (Serpin B9)	6	13	111
** I. Lysosomal Proteins **			
Laptm5 (Lysosomal membrane protein 5)	1938	6638	6180
Psap (Prosaposin glycosphingolipids)	(2039)	3559	12,296
Man2b1 (Alpha-mannosidase)	708	1621	5
Ctsb (Cathepsin B)	1175	4250	814
Ctsd (Cathepsin D)	3251	3595	1171
Ctsl (Cathepsin L)	451	2369	19
Ctsa (Cathepsin A)	(652)	2308	192
Ctss (Cathepsin S)	(1575)	1445	5290
Ctsz (Cathepsin Z)	360	571	1415
Ctsh (Cathepsin H)	(116)	316	329
Ctsc (Cathepsin C)	63	205	145
Ctso (Cathepsin O)	50	63	9
Ctsf (Cathepsin F)	48	53	1
Ctse (Cathepsin E)	21	76	0
** J. Immunoglobulin Receptors **			
FcgRIII (Fc gamma receptor 3)	773	1968	-
Fcgrt (FcRN)	484	1786	380
Fcgr4 (Fc gamma receptor 4)	56	82	-
Fcgr1 (Fc gamma receptor 1, high affinity)	17	35	51
Fcgr2b (Fc gamma receptor 2B, inhibiting)	(449)	9 *	59
Fcgr2a (Fc gamma receptor 2A)	-	-	580
Fcer1g (Fc-epsilon receptor gamma, signaling)	546	1318	1173
** K. MHC Classes I and II **			
B2m (beta-2 Microglobulin)	3358	5791	5521
H2-K1 (H2-K MHC Class I)	0 *	2606	-
H2-D1 (H2-D MHC Class I)	0 *	973	-
HLA-A	-	-	1548
HLA-B	-	-	6
HLA-C	-	-	2979
HLA-E	-	-	2984
HLA-DRB1	-	-	3023
HLA-DRA	-	-	5490
HLA-DPA1	-	-	2375
HLA-DPB1	-	-	1029
HLA-DPB2	-	-	0
HLA-DQB2	-	-	0
HLA-DQA1	-	-	103
HLA-DQA2	-	-	0
H2-DMa (H2-DM alpha chain)	0 *	53	-
H2-DMb2	0 *	9	-
H2-Aa (H2-IA)	0 *	52	-
** L. Classical Surface Receptors/Markers **			
CD14	560	627	1697
CD40	(25)	79	6
CD28	1	2	1
CD86 (B7-2)	63	257	236
CD80 (B7-1)	27	10	2
CD83 (Activation marker for dendritic cells)	(0.7)	0.1	18
CD244 (KIR2DL4)	29	34	31
CD84 (Ig superfamily, unknown function)	(244)	31	20
Mcemp1(Mast cell expressed membrane protein 1)	425	1140	0
CD209b (Receptor possibly involved in phagocytosis)	(227)	1125	4 (CD209)
CD209a	86	110	-
CD5l (CD5 like very specific for MQ, bind CD36)	558	892	0
Adgre1 (F4/80, Emr1, GPCR mucin like)	1178	373	87
Retnla (Relma, Fizz1, suppresses TH2 responses)	609	1223	30
** M. Scavenger Receptors **			
Marco (MARCO)	16	28	1
CD163 (Scavenger receptor, bind hemo-haptoglobin and complement)	9	30	225
CD36 (Scarb3) (Lung 1230)	100	562	250
CD68 (Binds oxidized LDL)	425	638	1273
CD177	9	15	0.3
Scara3 and 5	0	0	0
Scarb2	119	103	28
Scarb1	10	51	22
** N. Cytokine, Chemokine, and Endothelin Receptors **			
Fgfr1 (FGF receptor 1)	(279)	1429	0.1
Csf1r (M-CSF receptor)	678	1343	129
Csf2ra (GM-CSF receptor alpha chain)	277	838	47
Csf3r (G-CSF receptor CD114)	(73)	134	1236
Ccr5 (CCR-5 receptor)	76	119	2
Ccr1 (CCR-1 receptor)	228	91	34
Tnfrsf1b (TNF receptor Subfamily 1b)	66	399	1515
Tnfrsf1a (TNF receptor Subfamily 1a)	133	133	199
Tnfrsf11a (TNF receptor Subfamily 11a)	10	90	0.5
Tnfrsf21 (TNF receptor Subfamily 21)	28	80	25
Tnfrsf14 (TNF receptor Subfamily 14)	33	70	41
Il10ra (IL-10 receptor alpha)	(67)	280	609
Il15ra (IL-15 receptor alpha)	(8)	36	14
Il6ra (IL-6 receptor alpha)	(86)	34	80
Il4ra (IL-4 receptor alpha)	17	21	42
IL3ra (IL-3 receptor alpha)	21	17	8
Il13ra1 (Il-13 receptor alpha1)	13	10	123
EGFR (EGF receptor HER1)	(2	6	0
IL21r (IL-21 receptor)	5	5	0
IL27ra (IL-27 receptor alpha)	14	5	32
Il2rg (IL-2 receptor gamma)	(63)	4	47
Il2rb (IL-2 receptor beta)	1	3	1
Il1r1 (Receptor 1 for IL1 alpha)	(1)	1.4	2
Ednrb (Endothelin B receptor)	(1162)	1402	0.2
** O. Toll-Like Receptors and Accessory Proteins **			
Tlr4 (TLR-4)	29	200	24
Ly96 (MD2 LPS binding together with TLR4)	(74)	30	8
Tlr13 (TLR-13)	137	108	0?
Tlr7 (TLR-7)	(49)	84	9
Tlr1 (TLR-1)	(25)	71	6
Tlr8 (TLR-8)	(57)	65	46
Tlr2 (TLR-2)	43	26	40
Tlr3 (TLR-3)	8	18	0
Tlr6 (TLR-6)	18	3	2
Nlrc4 (Inflammasome related)	12	15	4
** P. Cell Adhesion **			
Itgam (Integrin alpha m, CD11b)	(2100)	11,106	245
Itga6 (Integrin alpha 6)	(751)	4519	0
Itgb2 (Integrin beta 2)	1602	3683	1975
Itgb1(Integrin beta 1)	718	1253	161
Itga4 (Integrin alpha 4)	86	393	230
Itgb7 (Integrin beta 7)	49	17	0?
Itgav (Integrin alpha v)	14	31	2
Itga9 (Integrin alpha 9)	12	39	2
Itgb3 (Integrin beta 3)	11	30	3
Itgax (Integrin alpha x, CD11c)	0.2	1	249
Selp (P-selectin)	679	3123	0
Emilin2 (Elastin microfibril located protein 2)	1123	2777	162
Icam2 (ICAM 2)	672	1092	39
Lgals3 (Galectin3, MAC2)	0	167	249
** Q. Chemokines and Cytokines **			
Pf4 (Platelet factor 4)	1437	3583	3
Ccl6 (Member of MIP-1 family)	1500	2616	-
Ccl9 (Also named MIP-1 gamma)	332	3013	0
Cxcl13 (B-cell attracting (BCA-1)	1253	1456	0
Ccl24 (Eotaxin-2 or MPIF-2)	480	707	1
Cxcl16 (T-cell and NK-cell attracting)	0	73	307
Cxcl14 (Attracting activated NK cells)	25	47	0
Cxcl2 (Also named MIP2 alpha)	42	22	12
Cxcl1 (Neutrophil attractant (Gro-a or NAP-3))	27	19	4
Cxcl12 (also named SDF1)	(30)	16	0
Ccl5 (Rantes attracts T-cells, Eosinophils and Basophils)	3	6	8
Ccl11 (Eotaxin 1)	0	0	0
Tgfb2 (TGF-beta 2)	(256)	1005	0.1
Tgfb1 (TGF-beta 1)	166	650	918
Il16 (IL-16)	19	71	0
Csf1 (M-CSF, expressed low in most tissues)	(8)	44	6
IL18 (IL-18)	18	38	58
Il18bp (IL-18 binding protein)	20	28	6
IL1a (IL-1 alpha)	14	31	0.2
Il15 (IL-15)	(6)	16	6
IL27 (IL-27)	3	3	7
Il13 (IL-13)	0	0	0
Il12a and b (IL-12a and b)	0	0	1 and 0
Tnf (TNF-alpha)	1	0.1	131
Igf1 (IGF-1)	11	80	0
Egf (EGF)	0.4	2	0
Pdgfa (PDGF-A)	5	10	0.4
Pdgfb (PDGF-B)	6	4	0.2
Vegfa (VEGF-A)	0	0	183
Vegfb (VEGF-B)	4	1.3	11
Vegfd (VEGF-D)	0	0	0?
** R. Signaling Components **			
Tyrobp (TYRO protein kinase-binding protein, Myeloid)	1504	2028	4617
Dab2 (Disabled homolog 2)	(146)	1413	1
Pde2a (cGMP-dependent cyclic phosphodiesterase)	(228)	1362	5
Slfn4 (Schlafen 4-myeloid signaling)	(361)	1322	0?
Btk	59	52	50
** S. Transcription Factors **			
Gata6 (GATA-6)	83	68	0
Gata3	0.3	0.6	0.3
Gata2	0.4	0.2	0.3
Gata1	0.1	0	0
Mitf	(15)	36	4
Spi1 (Pu.1)	228	536	1307
Myb	0.1	0	0.2
Runx1	(9)	139	42
Runx3	6	7	166
Creb3l1	2	2	0
Zeb2 (Zinc finger corepressor)	(100)	939	242
Tox2	46	78	0.3
Ikzf1 (Ikaros, Zinc finger transcription factor)	8	16	54
Foxp3	5	0.5	1

* Values that we are skeptical about and do not think they are correct due to the limitations of the particular technology as described in the text.

**Table 2 ijms-23-01185-t002:** Expression of pattern recognition- and angiogenesis-related proteins in mouse MΦs, mast cells and B-cells, and human monocytes. The number of reads for each of the different transcripts are given in actual numbers obtained from the Ampliseq analysis. To this table we also added the results from an Ampliseq analysis of unstimulated freshly isolated human peripheral blood monocytes from one individual (monocytes). The results from this individual and four additional individuals are shown in Appendix A. The B cells we analyzed are a clearly separate population of CD19 high and forward scatter low population of cells with medium–high CD11b (193 reads) and low CD5 (7 reads) levels of B cells, which based on these expression levels may represent B1b cells [24,25,26].

	Mouse Balb/c Mice			Human
	MΦs	Mast Cells	B-Cells	Monocytes
** A. TLR Rig-1 and MDA5 **				
Rig-1 (Ddx58)	27	44	104	3
MDA5 (Ifih1)	58	42	10	4
TLR-4	200	61	26	24
TLR-13	108	13	3	-
TLR-9	0.4	12	251	1
TLR-11	0	9	0	-
TLR-7	84	5	38	9
TLR-3 lung (41)	18	5	3	0
TLR-1	71	4	151	6
TLR-8	65	4	1	46
TLR-12	0.2	3	15	-
TLR-6	3	2	2	2
TLR-2	26	2	10	40
TLR-5 lung (10)	1	1	0	18
Dectin-1 (Clec7a)	261	11	2	142
Ccl5 lung (130)	6	43	15	8
** B. Angiogenesis related transcripts **				
Vegfa lung (350)	0	65	0	183
Vegfb	1	35	6	11
Vegfc	0	1	0	0
Vegfd lung (92)	0	1	0	-
Angpt1 (Angiopoetin 1)	0	72	0	0.4
Angpt2 (Angiopoetin 2)	0	0	0	0

**Table 3 ijms-23-01185-t003:** Transcript levels in the mouse liver. The analysis highlights genes that are highly expressed in liver, genes that are selectively expressed in liver, and genes of particular biological relevance for liver function. The number of reads for each of the different transcripts are given in actual numbers obtained from Ampliseq analysis. The same sample was analyzed twice, and the results from both analyses are depicted.

Ampliseq		
	Analysis 1	Analysis 2
** A. Major Liver-Specific Transcripts **		
Alb (Albumin, the major plasma protein)	59,900	61,827
Ashg (Alpha 2-HS glycoprotein/fetuin)	14,707	14,423
Hpx (Hemopexin bind heme)	5626	5520
Pzp (Pregnancy zone protein, alpha-2 globin family)	3975	3902
Ambp (Alpha-1-microglobulin)	3702	3830
Gnmt (Glycine-N-methyltransferase)	2386	2501
Vtn1 (Vitronectin)	2677	2721
Fn1 (Fibronectin)	925	1018
Hrg (Histidine-rich glycoprotein)	934	810
Akr1c6 (Aldo-keto reductase)	457	516
Akr1c14 (Alcohol dehydrogenase)	453	421
Tfr2 (Transferrin receptor 2)	439	451
Agt (Angiotensin precursor)	448	509
Hpn (Hepsin, a serine protease) (Kidney 370)	431	460
Tdo2 (Tryptophane 2,3-dioxygenase)	430	396
Afm (Afamin albumin related)	381	377
Sult2a2 (Sulfotransferase family 2A drug metabolism)	302	282
Sds (Serine dehydrase, serine metabolism)	292	303
Cp (Ceruloplasmin copper-carrying protein)	288	247
Dpys (Dihydropyrimidase pyrimidine metabolism)	272	270
Saa4 (Serum amyloid, apolipoprotein)	233	227
Apcs (Serum amyloid P component)	60	52
Msp1 (Macrophage stimulatory protein, also HLP)	323	251
Mup20 (Major urinary protein)	226	218
Cyp2c54 (Cytochrome P450 family 2 subfamily C)	225	228
Amdhd1 (Imidazolonepropionase histidine metabolism)	222	246
Ugt2a3 (UDP-glucuronosyltransferase 2A3)	218	217
Sult2a1 (Bile salt sulfotransferase)	176	172
Gckr (Glucokinase regulatory protein)	175	180
Fmo3 (Flavo containing mono-oxidase 3)	175	177
Asgr2 (Asialoglycoprotein receptor, galactose)	174	170
Baat (Bile acid-CoA amino acid N-acyl transferase)	172	159
Lyz2 (P-Lysozyme, probably Lyz1 instead)	160	155
Prodh2 (Hydroxyproline dehydrogenase)	144	146
Clec4f (Kupffer cell galactose receptor lectin)	121	115
Gfra1 (GDNF family receptor alpha 1)	111	113
Inhbc (Inhibin beta C-chain TGF-beta family)	100	103
Cpn2 (Carboxypeptidase N)	83	89
Gck (Glucokinase senses glucose levels)	83	87
Gys2 (Glycogene syntase)	79	82
Fgfr4 (FGF receptor 4)	58	54
Oit3 (Oncoprotein-induced transcript 3)	54	64
A1bg (Alpha-1-B glycoprotein)	40	46
Inhbe (Inhibin beta E chain precursor, TGF family)	39	43
Dnase2b (DNAse 2 beta)	37	41
Igfals (IGF-binding factor, stabilizes IGF in plasma)	31	33
Gdf2 (Bone morphogenic protein BMP-9)	30	29
Fgf21 (FGF-21 hepatokine, regulates sugar intake)	24	29
Il6ra (IL-6 receptor alpha)	21	21
Bmp5 (BMP-5, Bone morphogenic protein 5)	21	24
Thpo (Thrombopoietin regulates platelet production)	20	22
Bco1 (beta carotene metabolism)	20	20
Saa3 (Amyloid)	2.9	2.0
** B. Lipid Metabolism and Transport **		
Scd (Stearoyl CoA desaturase)	20,132	21,369
Fabp1 (Fatty-acid-binding protein)	15,683	16,162
Apoa1 (Apolipoprotein A1, major part of HDL)	15,093	15,512
Apoa2 (Apolipoprotein A1, part of HDL)	14,210	12,233
Apoc1 (Apolipoprotein C1, can be part of HDL)	12,723	12,195
Ttr (Transthyretin, transport thyroxin and retinol)	12,443	11,201
Gc (Gc-globin, vitamin D-binding protein)	9852	9765
Apoc3 (Apolipoprotein C3, can be part of VLDL)	9755	10,184
Rbp4 (Retinol-binding protein)	6483	6138
ApoE (Apolipoprotein E, transport lipids)	2445	2353
Sec14l4 (Sec14-like lipid binding 4, transport)	260	251
** C. Protease Inhibitors **		
Serpinc1 (Serpin C1)	4809	4845
Fetub (Fetuin b, Cystein protease inhibitor)	618	592
Itih1 (Inter alpha-trypsin inhibitor 1)	400	398
Itih3 (Inter alpha-trypsin inhibitor 3)	250	244
Serpina7 (Serpin A7)	42	46
** D. Coagulation Factors **		
Fgb) Fibrinogen beta)	7718	7474
Fga (Fibrinogen alpha)	4817	5144
Fgg (Fibrinogen gamma)	2540	2538
F2 (Thrombin)	3089	3507
F10 (Coagulation factor X)	1007	1144
Cpb2 (Carboxypeptidase B2, downregulates fibrinolysis)	664	641
F5 (Coagulation factor V)	522	527
Fgl1 (Fibrinogen-like protein 1)	452	476
F12 (Coagulation factor XII)	450	458
F13b (Coagulation factor XIII-B)	326	315
F7 (Coagulation factor VII)	150	142
F9 (Coagulation factor IX)	90	96
F11 (Coagulation factor XI)	57	49
F8 (Coagulation factor 8, Kidney (3 and 4), Uterus 16)	17	19
** E. Complement Factors **		
C3 (Complement factor 3)	5114	3902
Cfh (Complement factor H)	1231	1221
C4b (Complement factor 4B)	777	889
C4a (Complement factor 4A)	26 *	24 *
Cfi (Complement factor I)	530	472
Cfhr1 (Complement factor H-related protein)	167	172
C4bp (C4 binding protein regulatory)	735	671
C8g (Complement factor 8g)	393	398
Hc (Hemolytic component same as C5)	345	298
Crp (C-Reactive protein)	428	433
C1rl (C1r protease)	232	245
C1qbp (C1q binding protein)	209	203
C8b (Complement factor 8 beta chain)	127	119
C9 (Complement component 9)	119	114
C8a (Complement factor 8 alpha chain)	74	44
C1qb (Complement factor C1q beta chain)	52	31
C2 (Complement factor 2)	51	51
Fcna (Ficolin)	40	46
Cfp (Properdin)	30	31

* Indicates values that we are skeptical about and do not think they are correct due to the limitations of the particular technology as described in the text.

## Data Availability

All data is included in the manuscript.

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
