# Peer review of "Quantitative In-Depth Transcriptome Analysis Implicates Peritoneal Macrophages as Important Players in the Complement and Coagulation Systems"

_ijms, 2022, doi:10.3390/ijms23031185_

Round 1
Reviewer 1 Report
This study begins to address an interesting question regarding shared and distinct aspects of the transcriptomes of different mononuclear phagocyte populations. The focus of this study was a comparison of resident peritoneal macrophages from female Balb/C mice and human blood monocytes.
The central flaws are:
- There is no statistical analysis of the data and as presented it seems that each data set is based on a single replicate. Both biological and technical replicate with statistical analysis are required.
- Transcripts are parsed into categories, but no pathway analysis or other similar analyses of the data were performed. Rather, lists of genes are shown, but interetation and analysis are insufficient.
Author Response
Reviewer 1
- Biological and technical repetitions. Response- The mouse macrophage, B cell and mast cell values are based on material from 30 mice of a single mouse strain. In order to get sufficient number of cells we namely needed to wash the peritoneum of thirty mice for one experiment. It is therefore an average of 30 in-bread mice analyzed by two independent technologies RNA-seq and Ampliseq. The variation between individuals have thereby almost fully been removed and the mice are of the same age have the same living conditions why variations between individuals are at an absolute minimum. Repeating this type of analysis 4 times to get five samples would involve the sacrifice of 120 more mice, take at least six months due to renewal of ethical permit, and the result would most likely be almost identical. For the human monocytes we have four independent samples from five different individuals taken at different time points and of different sex. The values are so similar that we only show one of these samples as adding three additional samples would make the table almost unreadable. We have instead now added all the values from these five individuals in one supplementary table 1. Making average of these values would remove important quantitative information which we show is particularly important for the analysis of the MHC class I and II molecules and the Fc receptors which we now also comment in the article (Marked in red). For the liver we present two technical repetitions, analysis 1 and analysis 2 and as the reviewer can see the results are almost identical. Repeating this three additional times to get an almost identical value would involve several months of extra work, involve a lot of work and at a high cost with almost no additional information. I know that statistics sometimes can be very valuable when working with human samples, where the persons are of different age, different sex, different health condition, maybe taking drugs, different BMIs and the clinically important variations you want to look at are very small. However, in our case statistics on almost identical values gives no additional information but involves a massive amount of extra work the sacrifice of a large number of extra animals. 120 more mice for almost no additional information is in my world not defendable from an animal ethics perspective. We have now added the description of how the variation between individuals have been handled in this analysis by using material from 30 mice for a single analysis and that these mice have the same age, the same genome and also the same pathogen free living conditions.
- The use of pathway analysis. Response- Pathway analysis is a tool, similar to the heat maps, that most likely is designed by bioinformaticians with very little knowledge about biology. In the heatmap the data looks flashy with colors in different shades of blue and red. However, all quantitative information has been lost by this ¨kindergarten¨ type of presentation. To make data more reader friendly more than half of the information in the analysis have been sacrificed. It is maybe not exactly the same with the pathway analysis tool. However, I do not see how it could have been used in this study. We have focused on the biology of the macrophage and put it in the context of monocyte/macrophage biology and their role in immunity and tissue homeostasis. I think this is more important than a pathway analysis which I honestly have difficult to see what it would add to the story. I wish the reviewer could give me clues to what added value it would give and what pathways we should look at-Please give me a hint. We focus and discuss the complement and coagulation components, the difference between monocytes and macrophages in their role in tissue homeostasis, the difference in MHC class II expression and thereby role in antigen presentation, the high values of some but not other antibacterial components. The difference to other studies in the expression of some of the scavenger receptors. The difference in expression to other hematopoietic cell lineages in their expression of transcription factors and also the overall low expression of pattern recognition receptors. I therefore think that we discuss most of the most important issues connected to the new information originating from this study. I would have liked to have more information on the role of Fc receptor N on macrophages but we have no more information at the moment. There is also a lot of additional interesting findings in this material. However, we are afraid that the article would be too long and tiering for a reader to include also discussions around differences in proteases, protease inhibitors and several additional interesting differences between the peritoneal macrophages and the monocytes. However, this material is now by their presence in the tables available for a deeper analysis by us and other scientist in the field. However, we have added several additions to the discussion putting our results in the context of work from other labs and added a number of new references.
Reviewer 2 Report
I thank the authors for the presented article. The article contains unique data on the characteristics of peritoneal macrophages in comparison with other tissues containing resident macrophages, based on the analysis of the transcriptome. The information presented in the article is very relevant and its novelty is undeniable. The authors propose a new combined approach to the study of peritoneal mmacrophages. The presented study is unique in terms of the set of considered genes and transcription factors. Of course, your article deserves attention. However, the methods showed the use of female BALB / c mice. The number of animals participating in the study was not specified. There are no conditionally healthy donors whose monocytes from the mononuclear fraction of blood cells were analyzed (gender, age).
It is interesting why the authors chose for comparison circulating human monocytes, rather than mice.
Mouse abdominal macrophages are divided into two functionally different groups: large (CD11b hi F4 / 80 hi) and small peritoneal macrophages (CD11b + F4 / 80 lo). It is believed that CD11b + F4 / 80 lo are capable of presenting antigen to naive T lymphocytes (doi: 10.1038 / c 41598-018-33437-4). Why did the authors not take into account these subpopulations of peritoneal macrophages in their work?
The study also included B-lymphocytes isolated from the abdominal cavity. The author notes that a significant part of the cellular composition of the abdominal cavity is represented by macrophages and B-lymphocytes. B cells are considered control. The fact of the existence of B1 cells and classical B-lymphocytes (B2 cells), as well as the division of B1 cells into subpopulations B1a and B1b - lymphocytes (doi: 10.1111 / nyas.12760; doi: 10.1007 / 978-1-0716-1237-8 _2; DOI: 10.1007 / s 12185-019-02787-8). The method for detecting B-lymphocytes in this work was based on phenotyping cells for the CD19 molecule. Why were B cells examined using a classic surface marker? Peritoneal B1 lymphocytes are a unique pool of B lymphocytes expressing CD11b and CD5. Do you plan to take these subpopulations of B-lymphocytes into account in your future work? Perhaps the division of cells into smaller populations will help answer the questions that arise regarding the results obtained.
Why was it not possible to unify the methods of cell isolation? Use only flow cytometry and sorting? There are many articles that indicate activation of a mechanosensory signaling cascade that induces stress similar to that observed during inflammation (this applies in particular to peritoneal macrophages in mice) (doi: 10.1021 / acs.jproteome.8b00472).
In my opinion, if the discussion section is supplemented with a large number of references to the literature data described and the results obtained are analyzed in more detail, the article will become even more interesting for readers and attract more attention.
Author Response
Reviewer 2
- Thanks for many very good suggestions to improve the manuscript. We have tried to address them all.
- We have used the material from 30 female Balb/c mice of approximately the same age and living conditions. This information has now been added both to the Results and Materials and Methods sections (Marked in red).
- We have now added all the information concerning the transcriptome values for the human monocytes from five different blood donors in a supplementary table 1. This information has also been added to the text in the results and discussion sections. In this table we give age and gender of the donors.
- Why we did not use mouse monocytes. The number of monocytes is low in blood and we would have needed to kill a very large number of mice to get sufficient amount of blood and we had data from human monocytes from another project where we have analyzed LPS stimulation of monocytes and their role on their transcriptome. This is a large study we are working on and we thought the data from this study could be a very good complement to the macrophage story why we included the information from freshly isolated cells from these five donors as reference material to the mouse macrophages. The alternative would have been not to have such comparative analysis. This data was included late in the writing of the manuscript and we felt it clearly improved the manuscript. We get a few ml of blood from a mouse and for the purification of human monocytes we used the white blood cells from half a lite human blood. To get the same amount of mouse blood we may have to kill over 200 mice. A smaller number maybe sufficient but still a very large number of mice.
- We have now added information concerning the particular subtype of peritoneal macrophages throughout the entire manuscript (marked in red). We know both from the FACS picture and the transcriptome that it is the well-defined population of large peritoneal yolk sac derived macrophages and not the small monocyte derived macrophages.
- We have now added references and a short description of the B cells of this analysis. Based on CD11b and CD5 expression levels the most likely subpopulation is B1b cells as we also mention in the manuscript (marked in red). However, in my mind they appear to be too many to only be the B1b population as they are a smaller fraction of the peritoneal B cells but the trancriptome values clearly points in this direction.
- Why we used different methods for cell isolation. With FACS we had the possibility to get both macrophages and B cells at the same time and also have a good view of their phenotype and the different subpopulations all binding the same monoclonal antibodies. For the human monocytes MACS separation is much more convenient and used by our collaborator in routine studies why we selected these two methods as they were the most suitable for the different cell populations. MACS is very convenient when you do not need to have control over several similar subpopulations.
- We have now extended the discussion and included a number of new references to make it more interesting (Marked in red).
Reviewer 3 Report
This is a very good quality manuscript, the finding that the peritoneal MΦs, under non-inflammatory conditions, produce antimicrobial proteins such as lysozyme and complement components, and of several chemokines, including platelet factor 4 (PF4), Ccl6, Ccl9, Cxcl13 and Ccl24, and to express high levels of both TGFs, is very important. They also found that MΦs produce C1qA, C1qB, C1qC, properdin, C4a, factor H, ficolin, and coagulation factor FV.FX, FVII, and complement factor B. Comparison, of human peripheral blood monocytes. and mouse monocytes showed commonalities and differences The results of such comparison is important for many researchers working on rodent models and trying to translate the rodent results into human clinical appilcations.
Author Response
No modification or additions suggested by the reviewer.
Round 2
Reviewer 1 Report
Some essential details have been added to the Methods and other parts of the text in response to the Reviewers' comments. However, the fundamental weakness that was previously identified remains. Specifically, there is no statistical analysis of the data because the mouse peritoneal macrophage data are based entirely on a single biological replicate. The fact that cells from 30 mice were pooled to get this replicate does not make it data for 30 individual mice. The standard in the field is a minimum of 2 biological replicates and 2 technical replicates. This has not been met.
Second, inasmuch as the authors may believe that pathway analysis is childish, it is the current standard for unbiased data analysis. Ideally, it would be used to independently validate genes identified by the authors, but we this information was not provided.
Author Response
We have now revised the manuscript according to the Editors suggestion. We have added a section in the discussion describing the use of 30 mice to increase the amount of cells to get a good coverage for the quantitative analysis and also to obtain a good average value of the total transcriptome independent on individual variation. This new text has been marked in red in the revised manuscript.